# Could Long-Acting Cabotegravir-Rilpivirine Be the Future for All People Living with HIV? Response Based on Genotype Resistance Test from a Multicenter Italian Cohort

**DOI:** 10.3390/jpm12020188

**Published:** 2022-01-31

**Authors:** Andrea De Vito, Annarita Botta, Marco Berruti, Valeria Castelli, Vincenzo Lai, Chiara Cassol, Alessandro Lanari, Giulia Stella, Adrian Shallvari, Antonia Bezenchek, Antonio Di Biagio

**Affiliations:** 1Unit of Infectious Diseases, Department of Medical, Surgical, and Experimental Sciences, University of Sassari, 07100 Sassari, Italy; 2Infectious and Tropical Disease Unit, Department of Experimental and Clinical Medicine, University Hospital Careggi, University of Florence, 50134 Florence, Italy; annaritab1991@gmail.com; 3Department of Health Sciences (DISSAL), University of Genoa, 16132 Genoa, Italy; marco.berruti1@gmail.com; 4Infectious Diseases Unit, Ospedale Policlinico San Martino-IRCCS, Department of Health Sciences (DISSAL), University of Genoa, 16132 Genoa, Italy; antonio.dibiagio@hsanmartino.it; 5Department of Pathophysiology and Transplantation, University of Milano, 20126 Milano, Italy; valeria.castelli3@gmail.com; 6Infectious Disease Unit, Fondazione IRCCS Ca’ Granda Ospedale Maggiore Policlinico, 20122 Milano, Italy; 7Struttura Complessa di Microbiologia e Virologia, Dipartimento di Scienze Biomediche, Università di Sassari, 07100 Sassari, Italy; lai.vince@tiscali.it; 8Dipartimento Biotecnologie Mediche, Università degli Studi di Siena, 53100 Siena, Italy; chiaracassol92@gmail.com (C.C.); ale90-08@hotmail.it (A.L.); giulia.stella441@gmail.com (G.S.); 9UOC Malattie Infettive e Tropicali, AOU Senese, 53100 Siena, Italy; 10InformaPRO S.r.l., 00152 Rome, Italy; Adrian.Shallvari@informa.pro (A.S.); antonia.bezenchek@informa.pro (A.B.); 11EuResist Network GEIE, 00152 Rome, Italy

**Keywords:** antiretroviral treatment, cabotegravir, rilpivirine, long-acting, tailored treatment

## Abstract

Long-acting (LA) formulations have been designed to improve the quality of life of people with HIV (PWH) by maintaining virologic suppression. However, clinical trials have shown that patient selection is crucial. In fact, the HIV-1 resistance genotype test and the Body Mass Index of individual patients assume a predominant role in guiding the choice. Our work aimed to estimate the patients eligible for the new LA therapy with cabotegravir (CAB) + rilpivirine (RPV). We selected, from the Antiviral Response Cohort Analysis (ARCA) database, all PWH who had at least one follow-up in the last 24 months. We excluded patients with HBsAg positivity, evidence of non-nucleoside reverse transcriptase inhibitor (except K103N) and integrase inhibitor mutations, and with a detectable HIV-RNA (>50 copies/mL). Overall, 4103 patients are currently on follow-up in the ARCA, but the eligible patients totaled 1641 (39.9%). Among them, 1163 (70.9%) were males and 1399 were Caucasian (85.3%), of which 1291 (92%) were Italian born. The median length of HIV infection was 10.2 years (IQR 6.3–16.3) with a median nadir of CD4 cells/count of 238 (106–366) cells/mm^3^ and a median last available CD4 cells/count of 706 (509–944) cells/mm^3^. The majority of PWH were treated with a three-drug regimen (*n* = 1116, 68%). Among the 525 (30.3%) patients treated with two-drug regimens, 325 (18.1%) were treated with lamivudine (3TC) and dolutegravir (DTG) and only 84 (5.1%) with RPV and DTG. In conclusion, according to our snapshot, roughly 39.9% of virologically suppressed patients may be suitable candidates for long-acting CAB+RPV therapy. Therefore, based on our findings, many different variables should be taken into consideration to tailor the antiretroviral treatment according to different individual characteristics.

## 1. Introduction

Currently, HIV infection has become a chronic condition with a life expectancy comparable to uninfected people [1]. This is due to the introduction of modern antiretroviral therapy (ART) and its extensive use in both naïve and experienced patients, which leads to viral suppression in a high percentage of people with HIV (PWH) [2]. The treatment not only prevents the development of AIDS-defining pathologies but also reduces chronic inflammation and individual infectivity, thus reducing the risk of new infections [3,4]. 

Current ART regimens, particularly integrase inhibitors (INIs)-based ones, are characterized by high-efficacy, genetic barrier, and safety [5,6,7,8]. 

However, several downsides have emerged or remain critical: (i) the therapy is lifelong; (ii) the oral route is the only path of administration, leading to issues for patients with dysphagia or altered consciousness; (iii) all available compounds need to be taken daily, which requires high compliance by the patient; (iv) the aging of the HIV-population also leads to age-related pathologies (e.g., dementia) [9,10,11]; (v) polypharmacotherapy must account for the risk of incorrect drug assumption and interaction [12]; (vi) there are still patients with limited treatment options due to viral resistance for those in whom preserving available molecules remains critical [13,14]; and (vii) the current SARS-CoV-2 pandemic has led to a more complex and discontinued retention in care and treatment adherence [15]. Currently, clinicians should evaluate all the previously described characteristics to prescribe the perfect treatment for each person. 

In recent years, parenteral and long-acting (LA) formulations have been studied [16]. Indeed, LAs will contribute to achieving the treatment goals of ART: maximal and durable suppression of plasma viremia delays and prevention of the selection of drug-resistant mutations, preservation or improvement of CD4 T lymphocyte (CD4) cell numbers, and the conference of substantial clinical benefits. Moreover, it could represent an important alternative in the subset of patients with a lack of compliance to the oral therapy, such as women, non-Italian born, and intravenous drug users [4].

Recently, two trials investigated the LA parenteral combination of cabotegravir (CAB) (a second-generation INI), long-acting rilpivirine (RPV), and non-nucleoside reverse transcriptase inhibitors (NNRTI). They demonstrated non-inferiority to daily oral ART for maintaining HIV-RNA < 50 copies/mL in the setting of treatment-experienced patients (ATLAS) and treatment-naïve patients (FLAIR) with a once or twice a month administration schedule [17,18].

The long-acting CAB/RPV regimen was recently approved by the Food and Drug Administration (FDA) and the European Medicines Agency (EMA) and it will soon be available in Europe [19]. Long-acting CAB/RPV is administered every two months with two different intramuscular injections (IMI) [18,19].

Once long-acting CAB/RPV is available in Italy, infectious diseases ambulatory services will need to adapt to regularly administer long-acting CAB/RPV treatment to PWH, especially in the current COVID-19 pandemic setting; thereby, an estimation of the number and characteristics of the patients eligible for this treatment is needed.

The present study aims to analyze the number and characteristics of patients who would be eligible for treatment with long-acting CAB/RPV within an Italian PWH cohort.

## 2. Materials and Methods

### 2.1. Patients

We performed a retrospective analysis of the Antiviral Response Cohort Analysis (ARCA) database (https://db.dbarca.net/) to estimate the number of patients eligible for the novel therapy with long-acting CAB/RPV.

ARCA is a retrospective and prospective, longitudinal Italian cohort database created to monitor HIV-1 drug resistance and implement models to predict virological response to ART. ARCA currently includes >44,000 PWH followed at >50 clinical centers in Italy. Data collected include demographics, viral load (>630,000 records), HIV sequences (>55,000), CD4 counts (>740,000), treatments (>120,000), and AIDS-defining events (>8800). Thus, this database can be considered highly representative of the Italian cohort of PWH currently on ART. 

The inclusion criteria were being 18 years of age or older and having had at least one follow-up in the last two years. Regarding eligibility, we decided to adopt the same criteria used for the ATLAS and FLAIR trials [17,18]. Therefore, we considered eligible for treatment with long-acting CAB/RPV, PWH with an undetectable HIV-RNA (<50 copies/mL) for at least 12 months, who are HBsAg negative, and who do not have evidence of NNRTI (except K103N) or Integrase Strand Transfer Inhibitor (INSTI) mutations. Therefore, we considered not eligible all PWH with a detectable HIV-RNA, hepatitis B virus (HBV) coinfection, presence of NNRTI, or INSTI mutations. Furthermore, we excluded people with the following mutations for NNRTI: L100I, K101E/H/NP/Q, E138A/G/K/Q/R, V179L, Y181C/F/G/I/S/V, Y188L, G190A/C/E/Q/S/T/V, H221Y, F227C/L, and M230L. We also excluded PWH with the following INSTI mutations: T66I, E92Q, G118R, G140S, Y143A/C/G/H/K/R/S, S147G, Q148H/K/N/R, N155H/S/T, and R263K. 

Demographic (age, gender, risk factors for HIV transmission, and country of birth), clinical (HCV-antibody positivity, lower CD4 count (CD4 nadir), higher HIV-RNA, last CD4 cell count, HIV genotype resistance test (GRT), and HIV subtype), and treatment data (data of treatment start, current regimens) were retrieved from the database. 

### 2.2. HIV-1 Genotyping and Drug Resistance Evaluation

HIV-1 GRT was performed at each participating center based on local procedures and checked for consistency upon sequence uploading into the ARCA database. The database automatically extracts the mutations based on the HIV-1 consensus B sequence using a built-in local alignment script based on ClustalW and returns a 5-level susceptibility score based on the built-in AntiRetroScan algorithm. Pre-treatment Drug Resistance Associated Mutations (PDRAM) were defined as the detection of at least one mutation included in the World Health Organization (WHO)-recommended surveillance drug resistance mutations list and/or the International Antiviral Society-USA (IAS-USA) drug resistance mutations list.

### 2.3. Ethics

The ARCA cohort was approved by the Comitato Etico Regione Toscana Area Vasta Sud-Est (Regional Ethics Committee for Clinical Experimentation, Area Vasta South-East, Tuscany Region) with the ethic approval code ARCA/2014 of 21 July 2014. Written informed consent was obtained from all patients before participation. The study was performed following the ethical guidelines of the Declaration of Helsinki (7th revision) and the International Conference on Harmonization Good Clinical Practice guidelines.

### 2.4. Statistical Analysis

Quantitative variables were summarized as mean and standard deviation (SD) or median and 25°–75° percentiles (IQR), qualitative ones as absolute and relative (percentage) frequencies. The Shapiro–Wilk test was used to assess the normality of the data. Data analysis was carried out through STATA 16.1.

## 3. Results

Overall, of the 4103 patients currently on follow-up in the ARCA database, after excluding people with a detectable HIV-RNA, people with an HBV coinfection, and people without a GRT, 1883 PWH were selected. Furthermore, we excluded people having NNRTI or INSTI mutations (Figure 1). 

Therefore, 1641 (39.9%) patients met the eligibility criteria for treatment with long-acting CAB/RPV. The patients’ characteristics are summarized in Table 1.

Of the eligible patients, 1163 (70.9%) were males and 1399 were Caucasian (85.3%), of which 1291 (78.7%) were Italian born. Of 474 eligible women, about 12.2% were less than 50 years old and of childbearing potential. The median age of patients was 53 years (IQR 44–59). The most common routes of infection were sexual intercourse (1150, 70.1%) and intravenous drug use (252, 15.4%). HCV co-infection was present in 728 (44.4%) patients.

The median length of HIV infection was 10.2 years (IQR 6.3–16.3) and the median nadir of CD4 cells/count was 238 (106–366) cells/mm^3^. While the last available CD4 cells/count median was 706 (509–944) cells/mm^3^, only 48 (2.9%) of the participants had less than 200 CD4 cells/mm^3^. 

Regarding the HIV subtype, this data was present for 1576 (95.7%) PWH. The subtype B was prevalent (1174, 74.5%). Regarding subtype A1, it was detected in only 27 (1.7%) PWH. 

The majority of PWH were treated with a three-drug regimen (*n* = 1116, 68%); in particular, 568 (34.6%) PWH were treated with 2 NRTI + INI, 420 (25.6%) with 2 NRTI + NNRTI, and 128 (7.8%) with 2 NRTI and PI. Among the 525 (30.3%) patients treated with two-drug regiments, 325 (18.1%) were treated with lamivudine (3TC) and dolutegravir (DTG) and 84 (5.1%) were treated with RPV and DTG. The most common antiretroviral treatments are summarized in Table 2.

## 4. Discussion

Two open-label, phase III trials (ATLAS and FLAIR) demonstrated that long-acting CAB/RPV was non-inferior to oral therapy in terms of maintaining HIV-1 suppression in patients previously treated with oral antiretrovirals and in naïve patients. Applying the eligibility criteria proposed by these two clinical trials to the ARCA cohort, we found that about 39.9% may be switched to long-acting CAB/RPV [17,18]. Compared to the ATLAS and FLAIR trials, our patients were ten years older, mostly male, and Caucasian with a sexually acquired HIV infection. Less represented patient subsets were women and intravenous drug users (IDU). This is probably due to our restriction criteria of virological suppression not always being reached by IDUs who often lack adherence. However, mainly for their scarce compliance to the oral therapy, this subset of patients may be one of the main targets for this new therapy.

Regarding the current regimen, only 84 patients (5.1%) were already treated with RPV and DTG, and these patients would probably be the first people to switch to long-acting CAB/RPV. Additionally, PWH treated with 3TC/DTG without NNRTI resistance (325, 18.1% of our cohort) could easily be switched to the LA treatment. Together, the combination of three-drug regimens with INIs and NNRTI accounts for most ART represented in our dataset (34.6% and 25.6%, respectively). This is encouraging since long-acting CAB/RPV probably ensures a good response to a switch strategy that includes the same drug class. It should also be noted that in recent years, the proactive switch involves mainly these drug classes rather than PIs in order to limit their well-known toxicity [12,20,21].

One of the main problems regarding patients’ eligibility is the possible lack of GRT. In these cases, it would be mandatory to obtain the complete history of the HIV treatments, particularly if they have had an NNRTI or INSTI failure. In those PWH without GRT but with a previous virological failure or history of detectable HIV-RNA during treatment with NNRTI or INSTI failure, GRT from peripheral blood mononuclear cells (PBMCs) may be a valid alternative [22,23].

Once eligible patients are found, further considerations need to be addressed in order to identify the ideal candidate. First, since long-acting CAB/RPV needs to be administered in a hospital setting every two months, it will require at least six clinical visits annually; thus, it will be important to illustrate to patients the importance of keeping the ambulatory appointment every two months. Additionally, the distance from the center needs to be considered. However, in many centers, the hospital pharmacy distributes oral HIV treatment for two months, so there would be no difference in the number of hospital visits for PWH. Secondly, people with depression or mood disorders should be treated carefully because of the few data available about adverse effects on the central nervous system [17]. However, other INIs are associated with adverse neurological effects [24,25]. RPV may also cause depression exacerbation [18].

Thirdly, it will be important to perform an accurate pharmacological analysis because drugs that induce the UGT1A1 (uridine diphosphate glucuronosyltransferase family 1 member A1) and/or CYP3A4 (cytochrome P450 family 3 subfamily A member 4) [26] could decrease CAB and/or RPV concentrations, leading to loss of virologic response. Moreover, the body weight and composition need to be considered because people with a body mass index (BMI) > 30 kg/m^2^ show a higher risk of virological failure, probably due to slower CAB absorption [26]. Furthermore, some studies show that in obese people, there is a remarkable risk that the needle does not reach the muscle [27]. On the contrary, people with a low BMI could have a low tolerance to IMI, with a subsequent lack of adherence. In this regard, therapeutic drug monitoring (TDM), which is current practice for some antibiotics, antiepileptics, immunosuppressants, antifungals, and anti-HIV drugs, may play a role for a particular subgroup of patients, such as people with BMI > 30 kg/m^2^, people with drug interaction problems, or elderly people with sarcopenia [28]. Indeed, if these patients show a substantially lower or higher minimum blood plasma concentration, in order to improve efficacy, a personalized dosing schedule could be designed, with a consequent decrease of the costs [29]. Cutrell et al. demonstrated how virological failure in people treated with long-acting CAB/RPV had a multifactorial cause, but people with increased BMI, subtype A6/A1, and having at least two proviral RPV RAMs had an increased risk of failure [26]. Unfortunately, BMI is not collected in the ARCA database.

Moreover, we already know that this therapy may not be the best choice for women of childbearing potential (around 12.2% in our cohort) since many questions remain unanswered regarding the safety and efficacy of INIs and CAB administration during pregnancy. Recent studies suggest an unexpectedly low level of transplacental transfer of second-generation INIs (including CAB). Hence, the fetus may be protected from toxicity. However, it is not clear whether the reduced concentrations of CAB are sufficient to prevent mother-to-child transmission of HIV-1 as effectively as other regimens, or whether there may be a risk of selecting drug-resistant HIV-1 variants in the unborn child. These evaluations could limit the use of this class of drugs in women of childbearing potential, pending additional studies [30].

Finally, an important aspect that needs to be considered is the routine of ambulatory services because long-acting CAB/RPV cannot be self-administered. For this reason, administration will require several staff members designated to manage organization, perform IMIs, conduct appointments, and potentially reschedule missed IMI appointments. In addition, long-acting CAB/RPV needs to be stored between 2 and 8 ℃, which means the centers would need a dedicated method of refrigeration [19]. These aspects may have been challenging during the current SARS-CoV-2 pandemic as there was a temporary need to avoid non-urgent office visits and procedures in the ascending phase of infection. 

In this context, the use of LAs could represent both an issue and an opportunity. The closure of clinics in the case of a new wave of SARS-CoV-2 could make it impossible to perform the scheduled IMI. Furthermore, many centers in the recent period implemented telemedicine services to reduce the number of hospital visits, which could not be provided to patients being treated with long-acting CAB/RPV [31,32,33]. On the other hand, the administration schedule itself could limit access to outpatient services, including pharmacy services, therefore decreasing the risk of exposure to coronavirus. In addition, during the pandemic, a high number lost to follow-up were registered [34,35], which might have been avoided if those patients had scheduled visits for the IMIs. Finally, in the future we might look to organize home delivery of long-acting drugs, a strategy already adopted for other diseases to further reduce the number of hospital visits [33].

It must be noted that in the occurrence of a new recrudescence of the pandemic, organizing a dedicated ambulatory service or choosing an alternative solution, such as switching patients back to the oral route, should be based on local factors.

Our study presents some limitations. First, the data were extrapolated only retrospectively from an existing database designed to store genotypic data. Therefore, there is a lack of information such as stratification of the number of MSM and heterosexual transmissions, clinical information about patients’ comorbidities and baseline BMI, patients’ adherence to current therapies, or their desire to switch from oral to the parenteral route of drug administration. We also excluded patients with HBV coinfection or the absence of a TDF/TAF regimen. Pintado et al. argue that it is also essential to restrict the therapy to patients with a documented response to HBV vaccination (data not available in our cohort) since people with HIV are still at risk of HBV acquisition due to high-risk behavior and since HBV vaccination does not always elicit anti-HBs antibodies [36].

## 5. Conclusions

According to our study, about 39.9% of PWH currently on follow-up in the ARCA cohort may be suitable candidates for long-acting CAB-RPV. Based on our findings, infectious diseases ambulatory services will need to adapt accordingly, particularly in the current SARS-CoV-2 pandemic setting. Finally, when considering the switch of antiretroviral treatments, clinicians should evaluate clinical and demographical characteristics to find the perfect candidate for long-acting treatment.

## Figures and Tables

**Figure 1 jpm-12-00188-f001:**
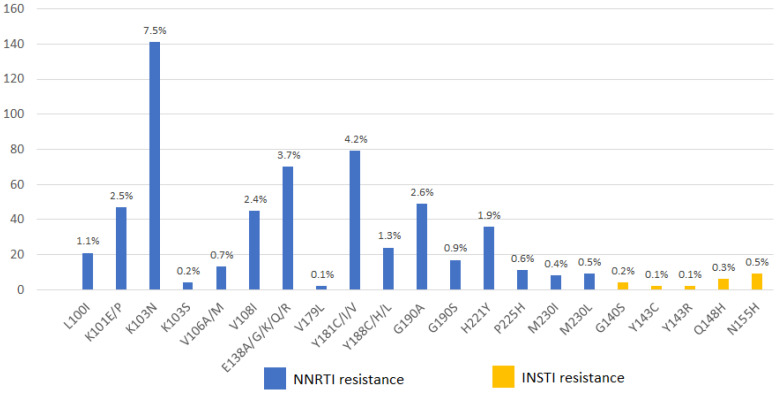
Distribution of non-nucleoside reverse transcriptase inhibitor (NNRTI) and Integrase Strand Transfer Inhibitor (INSTI) mutations in 1883 people living with HIV at baseline GRT in the ARCA cohort.

**Table 1 jpm-12-00188-t001:** Baseline characteristics of 1641 people with HIV eligible to cabotegravir + rilpivirine long-acting treatment.

Characteristics		N° of Patients
Age, Median (IQR) Years		53 (44–59)
Gender, *n* (%)	Male	1163 (70.9)
	Female	474 (28.9)
	Unknown	4 (0.2)
Ethnicity, *n* (%)	Caucasian	1399 (85.3)
	African	140 (8.5)
	Hispanic	58 (3.5)
	Asian	20 (1.2)
	Arabic	24 (1.5)
Route of transmission, *n* (%)	Sexual	1150 (70.1)
	IDU	252 (15.4)
	Vertical	7 (0.4)
	Transfusion	6 (0.3)
	Other	12 (0.7)
	Unknown	214 (13.1)
HCV coinfection, *n* (%)	Yes	728 (44.4)
	No	223 (13.6)
	Unknown	626 (3.8)
Length of HIV infection, median (IQR) years		10.2 (6.3–16.3)
Nadir CD4 cell count, median (IQR) cells/mm^3^		238 (106–366)
Nadir CD4 cell count < 200 cells/mm^3^, n (%)		677 (41.6)
Last CD4 cell count, median (IQR) cells/mm^3^		706 (509–944)
Last CD4 cell count < 200 cells/mm^3^, n (%)		48 (2.9)
HIV-RNA zenith, median (IQR) copies/mm^3^		80,330 (12,680–268,400)

IQR—interquartile range; IDU—injection drug user.

**Table 2 jpm-12-00188-t002:** Current antiretroviral regimens in 1641 people with HIV eligible for long-acting treatment with cabotegravir/rilpivirine.

	Antiretroviral Regimes	N° of People (%)
2NRTI + INSTI	ABC/3TC/DTG	173 (10.5)
	TAF/FTC/BIC	184 (11.2)
	TAF(TDF)/FTC + DTG	79 (4.8)
	TAF/FTC/EVG/c	38 (2.3)
	TAF(TDF)/FTC + RAL	84 (5.1)
	Other	10 (0.6)
2NRTI + PI	TAF(TDF)/FTC/DRV/c	61 (3.7)
	TAF(TDF)/FTC/DRV/r	17 (1.0)
	TAF(TDF)/FTC/ATV/r	15 (0.9)
	Other	35 (2.1)
2NRTI + NNRTI	TAF(TDF)/FTC/RPV	336 (20.5)
	TDF/3TC/DOR	17 (1.0)
	TDF/FTC/EFV	25 (1.5)
	Other	42 (2.6)
Dual Regimens	3TC/DTG	325 (19.8)
	RPV/DTG	84 (5.1)
	3TC + DRV/c	42 (2.6)
	DTG/DRV/c	13 (0.8)
	Other	61 (3.7)

NRTI—nucleoside analog reverse-transcriptase inhibitors; INSTI—integrase inhibitor; PI—protease inhibitor; NNRTI—non-nucleoside analog reverse-transcriptase inhibitors; ABC—abacavir; 3TC—lamivudine; DTG—dolutegravir; TAF—tenofovir alafenamide; FTC—emtricitabine; BIC—bictegravir; TDF—tenofovir disoproxil; EVG—elvitegravir; c—cobicistat; RAL—raltegravir; DRV—darunavir; r—ritonavir; ATV—atazanavir; RPV—rilpivirine; DOR—doravirine; and EFV—efavirenz.

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
