# Peer review of "Could Long-Acting Cabotegravir-Rilpivirine Be the Future for All People Living with HIV? Response Based on Genotype Resistance Test from a Multicenter Italian Cohort"

_jpm, 2022, doi:10.3390/jpm12020188_

Round 1
Reviewer 1 Report
De Vito et al. in their manuscript entitled, “Could Cabotegravir-Rilpivirin Long-Acting be the future for all people living with HIV? Response based on genotype resistance test from a multicenter Italian cohort”, find supportive evidence that the new CAB-RPV LA drug regimen would be beneficial to the HIV infected population in Italy. The authors examine the current HIV patient cohort in the ARCA database for the presence of mutations that would cause CAB-RPV LA treatment failure. Interestingly, the authors find that 39.9% of Italian HIV positive patients would benefit from CAB-RPV LA treatment, as they lack NNRTI and IN inhibitor mutations against CAB or RPV. However, the authors caution that since this therapy is an injectable, only patients that can easily access healthcare should be considered. Overall the paper provides a snapshot of the percentage of HIV infected patients in Italy that could benefit from CAB-RPV LA therapy.
Few comments:
1.) To be consistent for the reader, throughout the paper the treatment is referred to as “long-acting CAB+ LA-RPV” or “CAB-RPV LA” or “CAB and RPV-LA” or “RPV-LA” or “CAB-RPV-LA” or “long acting CAB-RPV” or “LA treatment”. Just pick one definition and abbreviation.
2.) Edits to wording/grammar/tense/sentence structure should be addressed throughout paper that may not have translated well. Suggest to have a native English person to read/edit paper. To name a few examples…
line 25: remove dash from “selec-tion”
line29-31: awkward sentence structure
line 48: led should be leads
line 49: I think the word “condition” should really be “treatment” or “therapy”
line 54: …”have” emerged…
line 63: remove the word “should”
line 230: “reducted” should be “reduced”
line 272: evaluated should be “evaluate”
etc….
3.) line 73: Define “INI” for the reader. And “INSTI” is also used interchangeably too. Maybe just use INSTI?. In Table 2 definitions, you state INSTI is integrase inhibitor, but it really stands for “integrase strand transfer inhibitor”.
4.) Line80: I think it is incorrectly stated that “CAB-RPV LA is administered every two months…”. It is actually 2 shots (one of CAB and one of RPV) given on the same day the patient goes to the clinic, and this occurs once a month. So the treatment instead of being oral for 365 days, is now lessened to 12 treatments (2 shots once a month, on the same day).
5.) Line 95 and 98 you use “PLWH” and no definition, versus everywhere else in paper it is PWH.
6.) Line 106: define “HBV”
7.) Line 107: define “*” as any amino acid substitution?
8.) Line140/Figure 1: Font is very small for the reader.
9.) I think Figure 1 is confusing. Is this the percentage of mutations from the 1,884 PWH before exclusion or the percentage of remaining mutations from the final 1,641 PWH? In lines 106-109 you state that for example people with L100I and G140S are excluded, but these mutations are in Figure 1?
Might be nice for the reader to show the comparison of % of mutations before and post exclusion to get a sense of which NNRTI and INSTI mutations are prevalent in the Italian population (even though this is influenced by the drug regimens used). The main interest of the paper is the pre-existing mutations in patients... and it should be highlighted more.
10.) Line 184: Define “3DR” for the reader
11.) Line 209: Need to add references and define for the reader what “UGT1A1” and “cYP3A4” are.
12.) Line 220: Define “Cmin” for the reader.
Author Response
Reviewer (R)1: De Vito et al. in their manuscript entitled, “Could Cabotegravir-Rilpivirin Long-Acting be the future for all people living with HIV? Response based on genotype resistance test from a multicenter Italian cohort”, find supportive evidence that the new CAB-RPV LA drug regimen would be beneficial to the HIV infected population in Italy. The authors examine the current HIV patient cohort in the ARCA database for the presence of mutations that would cause CAB-RPV LA treatment failure. Interestingly, the authors find that 39.9% of Italian HIV positive patients would benefit from CAB-RPV LA treatment, as they lack NNRTI and IN inhibitor mutations against CAB or RPV. However, the authors caution that since this therapy is an injectable, only patients that can easily access healthcare should be considered. Overall the paper provides a snapshot of the percentage of HIV infected patients in Italy that could benefit from CAB-RPV LA therapy.
Authors reply (AR): We would like to thank you for having revised our work and give us precious comment to improve ourh manuscript.
R: To be consistent for the reader, throughout the paper the treatment is referred to as “long-acting CAB+ LA-RPV” or “CAB-RPV LA” or “CAB and RPV-LA” or “RPV-LA” or “CAB-RPV-LA” or “long acting CAB-RPV” or “LA treatment”. Just pick one definition and abbreviation.
AR: We agree with you. We decided to use for the manuscript long-acting CAB/RPV.
R: Edits to wording/grammar/tense/sentence structure should be addressed throughout paper that may not have translated well. Suggest to have a native English person to read/edit paper. To name a few examples…
AR: We have re-read the manuscript and fixed the error present in the manuscrpt. Furthermore, according to your suggestion we have asked to a native English to edit the paper.
3.) line 73: Define “INI” for the reader. And “INSTI” is also used interchangeably too. Maybe just use INSTI?. In Table 2 definitions, you state INSTI is integrase inhibitor, but it really stands for “integrase strand transfer inhibitor”.
AR: We agree with you. We decided to use INSTI in the manuscript.
4.) Line80: I think it is incorrectly stated that “CAB-RPV LA is administered every two months…”. It is actually 2 shots (one of CAB and one of RPV) given on the same day the patient goes to the clinic, and this occurs once a month. So the treatment instead of being oral for 365 days, is now lessened to 12 treatments (2 shots once a month, on the same day).
AR: thank you for your comment. Cabotegravir-rilpivirin long-acting will be administered every two months with two shots at the same day, one of CAB and one of RPV. According to EMA, we decided to specify in the paper that there will be two different intramuscular injections at the same day.
5.) Line 95 and 98 you use “PLWH” and no definition, versus everywhere else in paper it is PWH.
AR: thank you for have carefully read the manuscript. We modified PLWH with PWH in the text.
6.) Line 106: define “HBV”
AR: Thank you for your comment. We added the definition of HBV (hepatitis B virus)
7.) Line 107: define “*” as any amino acid substitution?
AR: thank you for your comment. We used * as any aminoacid, but according to your comment we decided adding the different amino acid in orderd to make it clearer for the reader.
8.) Line140/Figure 1: Font is very small for the reader.
AR: Thank you for your comment. According to your suggention we used a bigger font.
9.) I think Figure 1 is confusing. Is this the percentage of mutations from the 1,884 PWH before exclusion or the percentage of remaining mutations from the final 1,641 PWH? In lines 106-109 you state that for example people with L100I and G140S are excluded, but these mutations are in Figure 1?
Might be nice for the reader to show the comparison of % of mutations before and post exclusion to get a sense of which NNRTI and INSTI mutations are prevalent in the Italian population (even though this is influenced by the drug regimens used). The main interest of the paper is the pre-existing mutations in patients... and it should be highlighted more.
AR: The figure 1 is the number of mutations of the 1883 PWH that have a HIV-RNA<50copies/mL and a negative HBsAg. People that will be eligble to be treated with long-acting CAB/RPV could not having this mutation. We specified it better in the text and in the caption of figure 1.
10.) Line 184: Define “3DR” for the reader
AR: thank you for your comment, we modified 3DR with 3 drug-regimens
11.) Line 209: Need to add references and define for the reader what “UGT1A1” and “cYP3A4” are.
AR: Thank you for your comment. We explained the significance of these acronyms in the text.
12.) Line 220: Define “Cmin” for the reader.
AR: we modified Cmin with minimum blood plasma concentration.
Reviewer 2 Report
In the current research article by De Vito et al. the authors have conducted a retrospective analysis to determine the eligibility of patients with HIV in ARCA Italian cohort for LA-drugs administration. Based on their analyses, majority of the PLWH seem to be not eligible for LA. I was curious why female were under-represented and what could be the reason for that? I appreciate that the authors have also discussed the limitations of the study in the manuscript as well as the requirements in the hospital settings for smooth administration of the LA drugs. Overall, it is a short but informative article for the field. I recommend to check for typographical errors as I found few while going through the text. For instance, resistance is incorrectly written in Fig 1.
Author Response
Reviewer (R)1: De Vito et al. in their manuscript entitled, “Could Cabotegravir-Rilpivirin Long-Acting be the future for all people living with HIV? Response based on genotype resistance test from a multicenter Italian cohort”, find supportive evidence that the new CAB-RPV LA drug regimen would be beneficial to the HIV infected population in Italy. The authors examine the current HIV patient cohort in the ARCA database for the presence of mutations that would cause CAB-RPV LA treatment failure. Interestingly, the authors find that 39.9% of Italian HIV positive patients would benefit from CAB-RPV LA treatment, as they lack NNRTI and IN inhibitor mutations against CAB or RPV. However, the authors caution that since this therapy is an injectable, only patients that can easily access healthcare should be considered. Overall the paper provides a snapshot of the percentage of HIV infected patients in Italy that could benefit from CAB-RPV LA therapy.
Authors reply (AR): We would like to thank you for revising our work and giving us precious comments to improve our manuscript.
R: To be consistent for the reader, throughout the paper the treatment is referred to as “long-acting CAB+ LA-RPV” or “CAB-RPV LA” or “CAB and RPV-LA” or “RPV-LA” or “CAB-RPV-LA” or “long acting CAB-RPV” or “LA treatment”. Just pick one definition and abbreviation.
AR: We agree with you. We decided to use for the manuscript long-acting CAB/RPV.
R: Edits to wording/grammar/tense/sentence structure should be addressed throughout paper that may not have translated well. Suggest to have a native English person to read/edit paper. To name a few examples…
AR: We have re-read the manuscript and fixed the error present in the manuscript. Furthermore, according to your suggestion, we have asked to a native English to edit the paper.
3.) line 73: Define “INI” for the reader. And “INSTI” is also used interchangeably too. Maybe just use INSTI?. In Table 2 definitions, you state INSTI is integrase inhibitor, but it really stands for “integrase strand transfer inhibitor”.
AR: We agree with you. We decided to use INSTI in the manuscript.
4.) Line80: I think it is incorrectly stated that “CAB-RPV LA is administered every two months…”. It is actually 2 shots (one of CAB and one of RPV) given on the same day the patient goes to the clinic, and this occurs once a month. So the treatment instead of being oral for 365 days, is now lessened to 12 treatments (2 shots once a month, on the same day).
AR: thank you for your comment. Cabotegravir-rilpivirine long-acting will be administered every two months with two shots at the same day, one of CAB and one of RPV. According to EMA, we decided to specify in the paper that there will be two different intramuscular injections on the same day.
5.) Line 95 and 98 you use “PLWH” and no definition, versus everywhere else in paper it is PWH.
AR: thank you for having carefully read the manuscript. We modified PLWH with PWH in the text.
6.) Line 106: define “HBV”
AR: Thank you for your comment. We added the definition of HBV.
7.) Line 107: define “*” as any amino acid substitution?
AR: thank you for your comment. We used * as any amino acid, but according to your comment, we decided to add the different amino acids in order to make it clearer for the reader.
8.) Line140/Figure 1: Font is very small for the reader.
AR: Thank you for your comment. According to your suggestion we used a bigger font.
9.) I think Figure 1 is confusing. Is this the percentage of mutations from the 1,884 PWH before exclusion or the percentage of remaining mutations from the final 1,641 PWH? In lines 106-109 you state that for example people with L100I and G140S are excluded, but these mutations are in Figure 1?
Might be nice for the reader to show the comparison of % of mutations before and post exclusion to get a sense of which NNRTI and INSTI mutations are prevalent in the Italian population (even though this is influenced by the drug regimens used). The main interest of the paper is the pre-existing mutations in patients... and it should be highlighted more.
AR: Figure 1 shows the number of mutations of the 1883 PWH that have an HIV-RNA<50copies/mL and a negative HBsAg. People who will be eligble to be treated with long-acting CAB/RPV could not have this mutation. Therefore, we specified it better in the text and the caption of figure 1.
10.) Line 184: Define “3DR” for the reader
AR: thank you for your comment; we modified 3DR with three drug-regimens
11.) Line 209: Need to add references and define for the reader what “UGT1A1” and “cYP3A4” are.
AR: Thank you for your comment. We explained the significance of these acronyms in the text.
12.) Line 220: Define “Cmin” for the reader.
AR: we modified Cmin with minimum blood plasma concentration.
Reviewer: In the current research article by De Vito et al. the authors have conducted a retrospective analysis to determine the eligibility of patients with HIV in ARCA Italian cohort for LA-drugs administration. Based on their analyses, majority of the PLWH seem to be not eligible for LA. I was curious why female were under-represented and what could be the reason for that? I appreciate that the authors have also discussed the limitations of the study in the manuscript as well as the requirements in the hospital settings for smooth administration of the LA drugs. Overall, it is a short but informative article for the field. I recommend to check for typographical errors as I found few while going through the text. For instance, resistance is incorrectly written in Fig 1.
Authors' reply: We would like to thank you for reading and commenting our study. Regarding the female, the percentage of eligible females is similar to the female non-eligible to be treated with long-acting CAB/RPV. This percentage reflects the percentage of females living with HIV in Italy.
Regarding the typographical errors, we asked a native English speaker to read and edit the manuscript. We hope that now it is clearer.